# Early Gestational Exposure to High-Molecular-Weight Phthalates and Its Association with 48-Month-Old Children’s Motor and Cognitive Scores

**DOI:** 10.3390/ijerph17218150

**Published:** 2020-11-04

**Authors:** Libni A. Torres-Olascoaga, Deborah Watkins, Lourdes Schnaas, John D. Meeker, Maritsa Solano-Gonzalez, Erika Osorio-Valencia, Karen E. Peterson, Martha María Tellez-Rojo, Marcela Tamayo-Ortiz

**Affiliations:** 1Center for Research on Nutrition and Health, National Institute of Public Health, Cuernavaca 62000, Mexico; libniavib@gmail.com (L.A.T.-O.); msolano@insp.mx (M.S.-G.); mmtellez@insp.mx (M.M.T.-R.); 2Department of Environmental Health Sciences, School of Public Health, University of Michigan, Ann Arbor, MI 48109, USA; debjwat@umich.edu (D.W.); karenep@umich.edu (K.E.P.); 3National Institute of Perinatology, Mexico City 11000, Mexico; lschnaas@hotmail.com (L.S.); erikaosorio4@hotmail.com (E.O.-V.); 4Department of Nutritional Sciences, School of Public Health, University of Michigan, Ann Arbor, MI 48109, USA; meekerj@umich.edu; 5Research Unit in Occupational Health, Mexican Social Security Institute, Mexico City 06720, Mexico

**Keywords:** phthalate, HMWP, pregnancy, trimester, neurodevelopment, MSCA

## Abstract

In utero phthalate exposure has been associated with neurodevelopmental disorders, nevertheless, trimester-specific susceptibility remains understudied. Our aim was to identify susceptible windows to the effects of gestational High-Molecular-Weight Phthalates (HMWP) exposure on 48 months’ neurodevelopment. We measured six HMWP metabolites (MEHP, MEHHP, MEOHP, MECPP, MBzP and MCPP) in urine samples collected during each trimester from women in the Early Life Exposure in Mexico to Environmental Toxicants (ELEMENT) cohort (*n* = 218). We assessed children’s motor (MS), cognitive (GCI) and memory (MeS) abilities using McCarthy Scales of Children’s Abilities (MSCA). We used linear regression models to examine associations between trimester-specific phthalate metabolites and MSCA scores, adjusted for sex, gestational age, breastfeeding, and maternal IQ. Although phthalate concentrations were similar across trimesters, first and second trimester phthalates were inversely associated with MS and GCI, with first trimester associations with MS being the strongest and statistically significant. Stronger associations were seen with MS and GCI among boys compared to girls, however interaction terms were not statistically significant. Our results suggest that early gestation is a sensitive window of exposure to HMWP for neurodevelopment, particularly in boys. Regulations on phthalate content in food as well as pregnancy consumption guidelines are necessary to protect future generations.

## 1. Introduction

Neurodevelopment is a critical life process, beginning in utero and ending in adolescence [1]. Identifying risk factors for neurodevelopment at the earliest possible life stages is of high public health relevance since long-term effects of neurodevelopment can impact an individual’s life, and, at a population level, can translate into economic loss for a country [2]. Gestational exposure to endocrine disruptors, such as phthalates has been increasingly associated with neurodevelopment.

Phthalates are ubiquitous chemical compounds found in PVC (polyvinyl chloride)-based products as plasticizer agents. These compounds are classified by their carbon chains, such as high-molecular-weight phthalates (HMWP), which are found in plastic containers, food storage bags, plastic packaging, ink labelling products, as well as PVC pipes used in the food processing industry [3]. Due to their unstable chemical nature they are easily transferred to the surfaces with which they come into contact, such as food [4]. In addition, phthalate migration to food has also been detected to increase with higher temperature and fatty food characteristics [5]. Concentrations of these compounds have been consistently identified in foods such as fruits, vegetables, red meat, chicken, and dairy, among others [6,7]. Furthermore, phthalate concentrations have been consistently identified in populations of different countries worldwide [8,9] including in children and women of reproductive age in Mexico [10,11].

Phthalates have been previously detected in amniotic fluid, therefore indicating their ability to penetrate the placental barrier [12,13]. Intrauterine phthalates appear to exert their action during sensitive developmental periods on the hippocampus and hypothalamus, as seen in animal models [14,15]. Evidence from epidemiologic studies relating prenatal phthalate exposure to different neurodevelopmental outcomes has grown in recent past years [16,17,18,19,20,21,22,23,24,25,26,27]. Phthalates have also been identified as endocrine disruptors [28,29,30], with sex having a differential impact on outcomes studied [31]. A recent meta-analysis found that more research is needed to evaluate the timing of exposure since many of the studies consider only one sample during pregnancy, as well as the differences in the associations by sex which have not been conclusive [19]. Hence, the aim of our study was to identify prenatal windows of vulnerability to HMWP exposure and their association with neurodevelopment, as well as the possible effect modification of these associations by sex in 48-month-old children of Mexico City.

## 2. Materials and Methods

### 2.1. Study Population

Our study population is a subset of mother-child pairs (*n* = 649) from the ELEMENT (“Early Life Exposure in México to Environmental Toxicants”) study for which children had a neurodevelopment assessment at 48 months of age. ELEMENT is an institutional collaboration between the Mexican National Institute of Public Health (INSP) in Cuernavaca, Morelos, Mexico and the University of Michigan in Ann Arbor, MI, USA [32]. Our analysis includes women from the second and third of three sequentially enrolled cohorts, who were recruited from 1997 to 2004 in hospitals serviced by the Mexican Institute of Social Security (IMSS) in Mexico City, Mexico, and followed in our study facilities from pregnancy through their offspring’s childhood. General inclusion criteria included having a ≤14 weeks’ pregnancy at the time of recruitment, and having plans to remain living in Mexico City for at least three years for follow-up. Exclusion criteria included having a high risk pregnancy. All participants signed an informed consent and data confidentiality was guaranteed. The study was conducted in accordance with the Declaration of Helsinki and study protocols were approved by the Institution Revision Board of each involved institution. For this study we required phthalate metabolites concentrations that were analyzed in archived maternal urine samples collected during pregnancy (*n* = 218).

### 2.2. Phthalates

Second void morning urine samples were collected at the first, second and third trimester of pregnancy visit, frozen at −80 °C and analyzed at NSF International (Ann Arbor, MI, USA). Six HMWP metabolites, comprising MEHP (mono(2-ethylhexyl) phthalate), MEHHP (Mono(2-ethyl-5-hydroxyhexyl) phthalate), MEOHP (mono(2-ethyl-5-oxohexyl) phthalate), MECPP (Mono(2-ethyl-5-carboxypentyl) phthalate), MBzP (monobenzylphthalate) and MCPP (mono-3-carboxypropyl phthalate) were measured in urine using isotope dilution–liquid chromatography–tandem mass spectrometry (ID–LC–MS/MS) as previously described [10]. Values below the limit of detection (LOD) were replaced with LOD/√2 [10]. Standardized procedures were applied for the calibration processes and reagent control. Phthalate metabolite concentrations were corrected by specific gravity to adjust for urine dilution variability, using the following formula: P_c_ = P[(SG_p_ − 1)/(SG_i_ − 1)], where P_c_ is the phthalate metabolite (ng/mL) concentration corrected by specific gravity, P is the measurement of the phthalate metabolite, SG_p_ is the urinary specific gravity median (first trimester = 1.017, second trimester = 1.014, third trimester = 1.0135), and SG_i_ is the urinary specific gravity for the participant sample per trimester.

Phthalates were classified into two groups based on molecular weight and typical sources of exposure: For the present analysis we focused on the High-Molecular-Weight Phthalates (HMWP) (ester side-chain lengths, five or more carbons) [33], the sum of all analyzed metabolites and whose main source is food packaging, containers, plastic tubing, and vinyl among others [33]; and DEHP (di-(2-ethylhexyl) phthalate) metabolites: the sum of MEHP, MEOHP, MEHHP and MECPP, with a main source from food [34]. We calculated the molar sum for each group by dividing each metabolite concentration by its molar mass, and adding the individual metabolite concentrations (μmol/L) subsequently. The geometric mean for each phthalate metabolite across pregnancy was also calculated for each mother using available trimester measurements.

### 2.3. Children’s Neurodevelopment

Neurodevelopment was assessed with the McCarthy Scales of Children’s Abilities (MSCA). This test consists of five subscales: verbal, quantitative, perceptual-performance, memory and motor.

The motor scale (MS) evaluates child’s fine and gross motor coordination. The first three subscales listed above integrate the General Cognitive Index (GCI), which has been considered an equivalent to intellectual coefficients evaluated by other psychological scales or tests. The memory scale (MeS) evaluates child’s immediate memory through visual and auditive stimuli.

Scales have the following distributions: MS and MeS, have a mean of 50 points and a standard deviation of 10 points. GCI has a mean of 100 points and a standard deviation of 16 points. The MSCA was administered by trained and standardized psychologists, who were blind to participant’s phthalate metabolite concentrations.

### 2.4. Covariates

Potential covariates in the study were selected based on previous literature [18,22,24,35], and included child’s sex, gestational age at birth, birth weight and length, breastfeeding, mother´s age, education and Intelligence Quotient (IQ). Child’s sex, gestational age at birth (weeks), birth weight and length were obtained from the hospital’s medical records. Breastfeeding (exclusive in weeks) and maternal age, education and IQ (measured with Wechsler Adult Intelligence Scale—WAIS), were obtained through applied questionnaires in our research facility. We excluded participants with pregnancies ≤32 gestational weeks. The Home Observation for Measurement of the Environment (HOME) (Standard score 0–45), obtained through a participant’s home visit was available for a subsample of 102 children. All variables were included as continuous in the different models.

### 2.5. Statistical Analysis

We first compared characteristics of the dyads included in our study to nonparticipants (children in the cohort who had a MSCA but no phthalate data). We then calculated measures of central tendency for all variables and used a Wilcoxon test for paired samples to evaluate differences between trimester-specific phthalate concentrations. Individual metabolites and their sums were log transformed to achieve normality. We analyzed the association for each MSCA scale separately using the total pregnancy geometric mean for each phthalate metabolite. Subsequently, we evaluated the association for phthalates during each specific trimester with MSCA outcomes in separate linear regression models. We tested the inclusion of covariates by assessing statistical significance or a >10% in the effect estimator. All models were adjusted for: infant sex, gestational age at birth, breastfeeding and maternal IQ. We created interaction terms between geometric means phthalates and for individual metabolites of each trimester and sex, and ran the separate models further adjusting for these terms. To assess effect modification, the main models were stratified by sex. Finally, we repeated the analyses including the HOME observation for the main models, and we stratified by sex for HMWP and DEHP sums. All statistical analyses were done using Stata Version 13.0., StataCorp LP, TX, USA.

## 3. Results

Characteristics of the 218 mother-child pairs comprising the analytic sample are shown in Table 1 and did not differ significantly from those who were not included in the study. Women were on average 26.8 (±5.7) years old, their mean education and IQ were 11.0 (±2.8) years and 92.1 (±20.9) points, respectively. The mean gestational age for children was 39.2 (±4.3) weeks, the mean birth weight was 3.2 (±0.4) kg, 46.8% were boys and the average length of exclusive breastfeeding was 8.1 (±5.9) weeks. The average scores for the MSCA were: GCI = 97.5 (±13.0), MS = 45.5 (±9.2) and MeS = 48.7 (±6.5). Finally, HOME average total score for the subsample of 102 participants was 35.2 (±6.6).

We were able to measure phthalates in 188 samples for the first trimester, 189 samples for the second trimester and 214 samples for the third trimester (Table 2). Comparing the results by trimester, MEHP and MCPP showed its highest levels on the first trimester, MEHHP, MEOHP and MECPP in the second trimester, and MBzP showed its highest levels in the third trimester. Statistically significant differences between trimesters were seen for MBzP between first and second trimester, first and third trimester, and between second and third trimester. Likewise, statistically significant differences were seen for MCPP between first and second trimester.

The results from the models using the pregnancy geometric means for the phthalate metabolites showed negative associations between all metabolites and their sums and the MS and GCI, however only those of HMWP, MEHHP, MEOHP, MECPP and DEHP in relation to the MS were statistically significant. Null associations were seen with the MeS (Appendix A: Linear associations between urinary phthalate metabolites pregnancy geometric mean and McCarthy Scales of Children’s Abilities).

Trimester specific analyses revealed that the effect estimate was stronger for the first trimester exposure, followed by the second trimester and null for the third trimester. The strongest effect estimate was mainly seen in association to MS.

Figure 1 shows the results of the first trimester exposure analyses, where we saw that all the individual metabolites and sums showed a negative association with MS, most being statistically significant except for MCPP. For the GCI, the associations were also negative for all metabolites and sums, although only MBzP was statistically significant. Associations for MeS were not consistent, neither statistically significant for any metabolite nor sums in any trimester (results not shown).

For the second trimester (Figure 2), the associations for the MS remained negative and statistically significant in general except for MEHP and MCPP. The associations with the GCI remained negative, except for MCPP, with no statistical significance for any metabolite or sum. Again, no associations were seen for the MeS. Models for the third trimester showed no associations with any scale (Figure 3).

The interaction term between the individual phthalates (using the geometric means across pregnancy, or trimester specific measures) and sex did not reach statistical significance (*p* < 0.1), however, this may be due to sample size. Hence, when assessing effect modification through analyses stratified by sex we saw different results depending on the trimester of exposure (no differences between boys and girls were seen using geometric means across pregnancy. Appendix A. McCarthy Scales of Children’s Abilities and individual geometric mean and sum of urinary levels of phthalate metabolites association by linear regression models, stratified by sex).

Boys showed stronger negative associations between first trimester phthalates and the MS and GCI scores compared to girls (Table 3). This difference was not seen in the second and third trimester exposures results (Appendix A. Adjusted regression coefficients for change in McCarthy Scales of Children’s Abilities associated with a second trimester ln-unit increase in urinary phthalate metabolite concentration, stratified by sex; and Appendix A. Adjusted regression coefficients for change in McCarthy Scales of Children’s Abilities associated with a third trimester ln-unit increase in urinary phthalate metabolite concentration, stratified by sex).

Finally, when conducting nonstratified trimester-specific analyses adjusting for HOME score, the association remained negative, was stronger and statistically significant for first trimester metabolites, such as MECPP. When stratifying by sex the association between first trimester HMWP and MS remained significant and negative but was twice as strong for boys (*n* = 32), and for girls (n = 35) remained negative but lost statistical significance. A similar effect estimate change was seen for first trimester DEHP model and MS.

## 4. Discussion

Our results documented an overall inverse association between the gestational exposure to high-molecular-weight phthalates and 48 months’ neurodevelopment, in particular with motor and cognitive development. We observed that boys had stronger inverse associations between HMWP and motor and general cognitive index scores. While we did not find important differences between phthalate concentrations across the three pregnancy trimesters, the first trimester of gestation was highlighted as a critical window of exposure.

We found a negative and significant association for the first trimester phthalates and most of the metabolites with MS, which was consistent for the second trimester. These results point at a motor impairment during fetal development, in line with previous findings in younger children [36,37]. Even though the association between prenatal exposure to phthalates and motor impairment across childhood has been controversial [38,39], recent evidence points at a consistent motor impairment at 11 years old, with one of our studied metabolites (MBzP) [40]. Further research is needed to evaluate the effects of DEHP metabolites in later stages of child development.

All coefficients for GCI scores were negative, supporting previous findings with similar negative association between DEHP metabolites and WISC-IV (Wechsler Intelligence Scale for Children) subscales [41]. Although our results did not reach statistical significance, this could be due to sample size, rather than a lack of an association, given the growing number of studies with similar results [22].

Our research team has previously reported sex differences in the association between third trimester exposure to phthalates and 24–36 month neurodevelopment, finding a negative association between DEHP phthalate metabolites and their sum, and girls Bayley´s Mental Index (BMI) [18]. In our study, although statistically significant results for GCI were found in the first trimester only with MBzP for boys, all coefficients were consistently negative for the rest of the metabolites. On the other hand, for the first trimester exposure and MS, a significant negative association was found for ∑HMWP, MEHP, MBzP and ∑DEHP for boys, and for ∑HMWP, MEHHP, MEOHP, MECPP and ∑DEHP for girls. Even though MEHP did not reach significance, the findings for girls highlight the importance of the first trimester window of exposure regarding DEHP, since MEHHP, MEOHP and MECPP are considered biomarkers of exposure [42]. When adjusting the first trimester models of MS for HOME, the effect modification for DEHP and HMWP seen on boys remained negative and significant (*p* < 0.05), and the effect estimate was twice as strong, however, these results should be taken cautiously due to sample size.

According to worldwide distribution of three out of six phthalate metabolites analyzed in our study median urinary phthalate levels of MEHP in our population were above those found in the United States of America, The Netherlands, Israel and Japan (Appendix A: Median urinary phthalate metabolites levels during pregnancy, measured in different studies). MBzP levels found in our study were above those of Japan, Taiwan and Brazil, and below those found in France, United States of America, The Netherlands, Israel and Japan. Finally, MCPP levels in our study were above those found in the Netherlands Brazil and Israel. On the other hand, MCPP levels in France and the United States of America were higher than those found in our study. However, since MCPP and MBzP showed a difference between trimesters, results regarding these metabolites should be taken cautiously.

Despite rapid human urinary elimination of phthalate compounds, women in our study had detectable levels of several metabolites during pregnancy. In particular, noticeable elevated levels of DEHP compounds were seen, in line with previous findings of phthalate concentrations in Mexican food cans and plastic food containers [11]. These metabolites have been also previously found in several food groups such as bread, dairy, oils and fats, nuts, meat products, cereal, fish, sugar and preserves, poultry, carcass meat and other vegetables [43]. Given the usual source of exposure through the food process industry, further research is needed to establish if ultra-processed food consumption could account for a possible source of exposure in our population, as recent findings from studies point out [44]. Still, phthalate metabolite levels in the Mexican diet must be determined, as well as daily intake per age group, in order to target effectively the exposure hazard through public health policies.

The biological mechanisms proposed through experimental models, in which phthalates could affect embryonic development include a neuronal degeneration caused by disruption of cellular ionic homeostasis mediated by the hypoactivity of the membrane Na^+^ /K^+^-ATPase, which selectively affects structures that undergo marked, rapid structural and plasticity changes rapidly during development, such as the hippocampus [15]. Other potential pathways by which phthalates may influence neurodevelopment include thyroid homeostasis disruption [26], undifferentiated neuron apoptosis caused by peroxisome proliferator-activated receptors (PPAR) activation [45], fetal brain lipid profile alterations [46], and antiandrogenic activity [47].

As pointed out before, during embryonic and fetal development there are critical periods of susceptibility, in which toxic agents or substances can act and produce neurodevelopment alterations. However, these alterations may not be evident at birth, but can manifest themselves in the postnatal development stages [48]. Recent evidence in experimental animal models support that DEHP exposure in critical periods of fetal development may alter hypothalamic gene programming permanently, impairing somatic and reproductive development in male rats [14].

Our study has several strengths, such as its longitudinal design. This is among the few studies that evaluate associations between three trimester specific exposures and motor skills. Since our analysis included urinary phthalate metabolite samples from each pregnancy trimester, it is likely a more accurate reflection of exposure across pregnancy compared to one phthalate measurement. This aids to decrease the possible measurement bias in similar studies. However, multiple sampling per trimester could contribute to a better approach to phthalate exposure, as one measurement per trimester may not characterize it, being, therefore, a limitation or our study.

The main limitation of our study was the number of women for whom we had archived urinary samples to measure phthalate metabolites, most likely limiting our power to detect interactions with sex. Nevertheless, we observed stronger associations for boys than girls. Although seizures, hearing impairment and severe visual problems in children, are impairments that can affect neurodevelopment and thus can bias the findings, we did not assess them. However, if a participant was considered to have a condition that could interfere with the correct test application, the test was not applied, or comments were included in the database and considered in the data analysis. We were unable to account for maternal gestational stress or depression, which are known risk factors for neurodevelopment that can interact with chemical exposures [49]. Likewise, we did not collect information for trauma or abuse during infancy. We focused on high-molecular-weight phthalates, which are most common in food items; however, associations between low molecular weight phthalates and neurodevelopment have also been reported.

## 5. Conclusions

Our study contributes to the evidence of sensitive windows of exposure to HMWP phthalates, highlighting the need of preventing gestational exposure mainly in the first and second trimester. Likewise, our study supports the need of guidelines for toxic awareness consumption during pregnancy, in order to protect vulnerable groups such as children and women of child bearing age.

## Figures and Tables

**Figure 1 ijerph-17-08150-f001:**
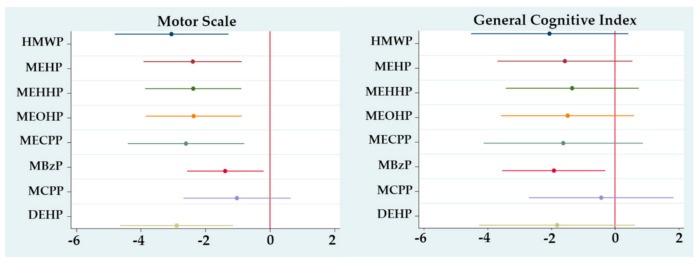
Regression coefficients for change in McCarthy Scales of Children’s Abilities (MSCA) associated with a first trimester ln-unit increase in urinary phthalate metabolite concentration, adjusted for sex, gestational age, breastfeeding and maternal IQ. **Motor Scale: only MCPP p > 0.05; General Cognitive Index: only MBzP p < 0.05.*

**Figure 2 ijerph-17-08150-f002:**
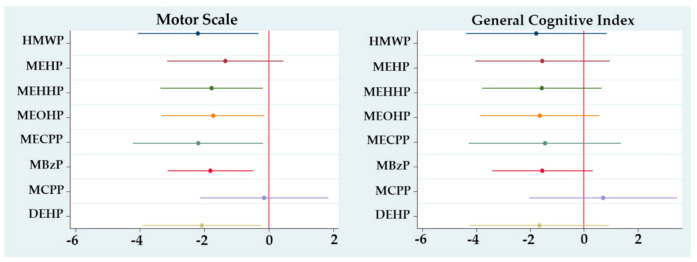
Regression coefficients for change in McCarthy Scales of Children’s Abilities (MSCA) associated with a second trimester ln-unit increase in urinary phthalate metabolite concentration, adjusted for sex, gestational age, breastfeeding and maternal IQ. **Motor Scale: MEHP and MCPP p > 0.05; General Cognitive Index: all metabolites p > 0.05.*

**Figure 3 ijerph-17-08150-f003:**
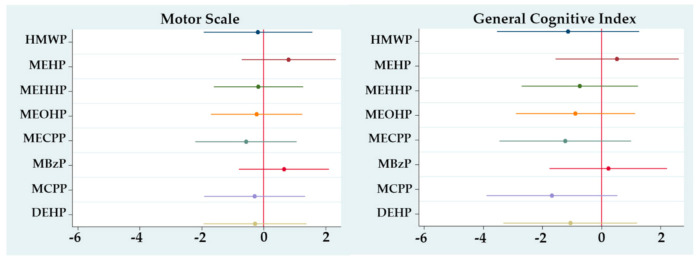
Regression coefficients for change in McCarthy Scales of Children’s Abilities (MSCA) associated with a third trimester ln-unit increase in urinary phthalate metabolite concentration, adjusted for sex, gestational age, breastfeeding and maternal IQ. **Motor Scale: all metabolites*
*p > 0.05; General Cognitive Index: all metabolites p >*
*0.05.*

**Table 1 ijerph-17-08150-t001:** Sociodemographic characteristics of the study mother-child pairs.^.^

Characteristics	Participants *n* = 218	Nonparticipants * *n* = 431
**Mother**	(mean ± SD)	(mean ± SD)
Age(years)	26.8 ± 5.7	26.5 ± 5.3
Education (years)	11.0 ± 2.8	10.0 ± 2.2
Maternal IQ	92.1 ± 20.9	88.6 ± 20.7 ^a^
**Child**		
Female (%)	53.2	49.4
Gestational Age (weeks)	39.2 ± 4.3	38.7 ± 4.3
Anthropometry		
Birth Weight (kg)	3.2 ± 0.4	3.1 ± 0.4
Birth Length (cm)	50.2 ± 2.0	49.7± 3.5 ^b^
Exclusive Breastfeeding (weeks)	8.1 ± 5.9	7.7 ± 5.8 ^c^
McCarthy Scales of Children’s Abilities Scores ^1^		
General Cognitive Index (GCI)	97.5 ± 13.0	97.2 ± 13.7
Motor Scale (MS)	45.5 ± 9.2	46.1 ± 9.9
Memory Scale (MeS)	48.7 ± 6.5	46.6 ± 7.1
HOME ^2^	35.2 ± 6.6 ^d^	35.8 ± 5.6 ^e^

^a^ (n = 397); ^b^ (n = 428); ^c^ (n = 188); ^d^ (n = 102); ^e^ (n = 20); ^1^ Standard Score (mean ± SD): General Cognitive Index (100 ± 16); Motor scale and Memory Scale (50 ± 10); ^2^ Home Observation for Measurement of the Environment. Standard Score (0–45); * No significant differences were observed between participants and nonparticipants.

**Table 2 ijerph-17-08150-t002:** Urinary phthalate metabolites geometric mean levels per trimester, adjusted for specific gravity.

Metabolite (ng/L)	First Trimester *n* = 188 (Mean ± SD)	Second Trimester *n* = 189 (Mean ± SD)	Third Trimester *n* = 214 (Mean ± SD)
High Molecular Weight (HMWP)			
MEHP	9.5 ± 12.5	7.8 ± 10.6	8.4 ± 11.8
MEHHP	30.5 ± 30.3	32.5 ± 72.1	30.5 ± 31.2
MEOHP	16.3 ± 15.8	19.4 ± 41.4	18.4 ± 19.0
MECPP	48.6 ± 42.3	55.4 ± 144.0	46.7 ± 43.0
MBzP *	6.6 ± 10.1	4.6 ± 5.6	7.3 ± 10.7
MCPP **	2.4 ± 7.4	1.6 ± 1.4	1.7 ± 1.8

* MBzP *p* < 0.05 between first and second trimester, first and third trimester, and between second and third trimester; ** *p* < 0.05 between first and second trimester.

**Table 3 ijerph-17-08150-t003:** Regression coefficients^†^ for first trimester ln-unit increase in urinary phthalate metabolite concentration associated with McCarthy Scales of Children’s Abilities, stratified by sex ^†^.

	General Cognitive Index		Motor Scale	
	Boys *n* = 77	Girls *n* = 90	Boys *n* = 77	Girls *n* = 90
**Metabolite**	ß ^†^ (95% CI)	ß ^†^ (95% CI)	ß ^†^ (95% CI)	ß ^†^ (95% CI)
HMWP ^1^	−3.4 (−7.1, 0.3)	−0.6 (−4.1, 2.9)	−3.2 * (−6.1, −0.4)	−2.5 * (−4.9, −0.2)
MEHP	−3.1 (−6.2, 0.1)	−0.1 (−3.1, 3.0)	−3.0 * (−5.4, −0.5)	−1.7 (−3.8, 0.3)
MEHHP	−2.1 (−5.0, 0.9)	−0.2 (−3.3, 2.9)	−2.2 (−4.5, 0.1)	−2.2 * (−4.3, −0.1)
MEOHP	−2.2 (−5.2, 0.8)	−0.4 (−3.5, 2.7)	−2.3 (−4.6, −0.0)	−2.1 * (−4.2, −0.0)
MECPP	−2.7 (−6.5, 0.9)	−0.2 (−3.8, 3.4)	−2.4 (−5.3, 0.5)	−2.4 * (−4.8, −0.0)
MBzP	−2.5 * (−4.9, −0.1)	−1.3 (−3.6, 1.0)	−2.3 * (−4.2, −0.5)	−0.5 (−2.0, 1.1)
MCPP	−1.9 (−5.9, 2.2)	0.3 (−2.5, 3.2)	−1.9 (−5.1, 1.2)	−0.5 (−2.4, 1.5)
∑ DEHP ^2^	−3.1 (−6.8, 0.5)	−0.3 (−3.8, 3.2)	−2.9 * (−5.7, −0.1)	−2.5 * (−4.8, −0.2)

**^†^** Adjusted for gestational age, breastfeeding and maternal IQ. ^1^ High-Molecular-Weight Phthalate Molar Sum: MEHP, MEHHP, MEOHP, MECPP, MBzP and MCPP. ^2^ Molar Sum of the four metabolites derived from DEHP: MEHP, MEHHP, MEOHP and MECPP. * *p* < 0.05.

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
