# Peer review of "Early Gestational Exposure to High-Molecular-Weight Phthalates and Its Association with 48-Month-Old Children’s Motor and Cognitive Scores"

_ijerph, 2020, doi:10.3390/ijerph17218150_

Round 1

Reviewer 1 Report

Thank you for the opportunity to review this paper. In this interesting manuscript, authors demonstrated the analysis of mother's exposure  to phthalates and its associations with children's results.

In my opinion, this is an interesting topic. The manuscript is well written.

My note:

Introduction

Although the introduction introduces enough in the subject, please change the order of the information.

1.Please start with a paragraph with general information on the important role of neurodevelopment at an early age (as in the sentence in Line 48) and on the effects of neurodevelopmental disorders on further life (and for public health). And please write here a general information that it is important to identify early risk factors for neurodevelopmental disorders.

  1. Next should be a paragraph from Lines 37-47.
  2. The next paragraph should include text from lines 49-56. Please highlight here what are the reasons for your investigation. Whether the results so far (available in the literature) are divergent, whether there is little research, or whether the sample sizes in these studies are small. Were the gender-related results statistically significant or divergent.
  3. The last paragraph is Lines: 56-58

Results

Table 1. Please indicate in this Table that the results did not differ significantly statistically between the groups (e.g. „Non patricipants *”)

Table 2. Please make a brief description under the table. And in the Table, please indicate the statistically significant results for MBzP * , and for MCPP **

Figures 1-3. Please write under each Figure that the results were not statistically significant, or write which result was statistically significant.

Discussion

Please change the order of paragraphs:

1.The first paragraph is good. Please add a short note about the results for the sex of the fetus.

  1. The following paragraphs should include, in turn, texts from the Lines: 205-213; 214-218, 225-237; 245-254; 255-265.
  2. Then, please write about the possible mechanisms of found relationships, Lines 197-204; 219-224.
  3. Then, please list the advantages of the test. Please list the restrictions separately.

Kind regards

Reviewer 2 Report

Firstly i would like to thank the authors for their hard work and excellent paper.

My only valid criticism is in the preparation of tables and figures which both need some improvement. Tables can be better formatted to make lines across easier to read and must be consistent with their presentation of mean SD in either left hand column of categories. Figure legends need to be below the figure and are of a different font. Figures should be redrawn to be clearer to the reader as they are extremely small.

Reviewer 3 Report

The prospective study examines the association between in-utero phthalate exposure during three trimesters on 48-months neurodevelopment. The findings are important and adds knowledge to further understanding of sensitive windows of exposure to phthalate. The manuscript is well written. I have few comments to make:

  1. If authors assessed for presence of seizures, hearing impairment, severe visual problems in children, as these impairments can affect neurodevelopment and thus can bias the findings.
  2. Did authors collect data on mother’s perinatal depression, anxiety or affective disorders which have evidence to affect developing babies’ neurodevelopment.
  3. Post-partum depression, trauma/abuse during infancy/childhood can also affect neurodevelopment.
  4. If pregnant mothers were taking any other medications (both prescribed or over the counter).
  5. If authors assessed for level of stress (interpersonal, job, living and financial related) during pregnancy and after-birth, as stress can affect neurodevelopment.
  6. Authors don't seem to have enlisted 'limitations' of the study.
